# Non-Linearity of Thermosetting Polymers’ and GRPs’ Thermal Expanding: Experimental Study and Modeling

**DOI:** 10.3390/polym14204281

**Published:** 2022-10-12

**Authors:** Alexander Korolev, Maxim Mishnev, Dmitrii Vladimirovich Ulrikh

**Affiliations:** 1Department of Building Construction and Structures, South Ural State University, 454080 Chelyabinsk, Russia; 2Department of Town Planning, Engineering Systems, and Networks, South Ural State University, 454080 Chelyabinsk, Russia

**Keywords:** thermal aging, polymer composites, glass fiber-reinforced plastics, coefficient of thermal expansion, thermo-relaxation, representative volume element, finite element method

## Abstract

Thermal expanding is the important property that defines the stress–strain condition of GRP structures exploited under heating and having limited thermal resistance. So, the GRPs’ thermal expanding prediction is the actual requirement of such structures design. The experimental accurate dilatometric study resulted in the non-linearity of thermosetting polymers and plastics thermal expanding under heating. The polymers and plastics thermal expanding coefficient (CTE) is non-linearly increasing under heating before glassing temperature (Tg). Using the previous polymers and GRPs modelling experience and experimental dilatometric results, the non-linear adequate prediction models of their CTE were proposed and proved. The new compensative wave model of polymers’ CTE and multi-layer model of GRPs’ CTE were proposed and successfully tested. A prediction of the temperature dependences of the thermal expansion coefficients of various thermoset polymer binders and data on the reinforcement structure was performed based on the experimentally obtained temperature dependences of the CTEs of GRPs. The prediction was performed using the finite-element homogenization method in the Material Designer module of the academic version of the Ansys package. A satisfactory concurrence of the numerical results of the prognosis and the experiment for all considered cases is observed in the temperature range from 50 to 100 °C, after glass transition temperature best coincidence of numerical values of CTE is obtained for glass-reinforced plastics on epoxy resin, which were not subjected to thermal aging.

## 1. Introduction

The developing of various composite structures subjected to transient thermomechanical actions demands a tool for evaluating the thermal expansion of materials at the stage of selecting components and composite design. The above-mentioned structures include, for example, fiberglass structures of gas exhaust ducts [1,2,3]; they can be operated for a long time at elevated temperatures, including those exceeding the glass transition temperature, which leads to changes in several properties [4,5,6,7]. Thermal expanding (TE) is the significant factor of thermal stresses developing in polymer and GRP structures during exploitation. In gas chimneys, the temperature gradient can achieve 100 °C and more. Taking that into account, the design of GRP structures demands accurate prognosis of CTE in a definite temperature range. Thus, the polymers’ and GRPs’ TE modeling that provide prediction models of the physical and mechanical properties of composites, based on the known characteristics of its components and the structure formed by them, is the actual problem.

Most of the research performed in this area could be classified on thermal expanding studies of compounds [8,9,10,11] and composites such as GRPs and hybrid [12,13,14,15,16,17,18,19] materials.

Thermal expanding is usual for glassed polymers and characterized by CTE, but much research points to the relationship between CTE and heating temperature, i.e., the non-linearity of polymers’ thermal expansion under heating [9,11,20]. In most cases, it was related with the polymer molecules conformation mechanisms, 3D molecule replacement and the distribution. This alteration causes the change and non-linearity of CTE [21,22,23].

Thermal expansion of polymer composites and hybrids is more complex due to the influence of all components on the summary expansion result. Different authors propose different prognosis models consisted of two or three phases with Van der Waals interphase bonds in the inter-transition zone “matrix-reinforcing element” (ITZ) [18,24]. However, these models are not realized in the non-linearity of polymers’ and composites’ thermal expanding. It is accurate that the inclusion of components with less CTE (mineral additive, graphene tubes, glass, and basalt fibers) results in less CTE of the composite [19,25,26,27,28,29,30,31]. Thus, the greater the linear thermal expansion contribution of the components characterized by the coefficient of linear thermal expansion (CTE), the greater the linearity of the thermal expansion of the composites. It is necessary to consider principal difference in thermal expansion under heating before and after glassing temperature (Tg) [17,32,33]. Additionally, there is almost no research on glassed polymers’ thermal expansion after prolonged thermal relaxation under temperatures above Tg.

The most highlighted research achieved just empirical models of polymers and composites thermal expansion. Previous authors’ experience of modelling polymers’ and composites’ deformability proved that the most adequate are supra molecular models presented as a domain structure bonded by surface bonds with changing stiffness under heating [33,34].

The prediction of the CTE of glass-reinforced plastics (GRP) along with elastic characteristics as a function of temperature can be performed using multiscale modeling and finite-element homogenization [35,36]. There are also analytical methods, for example, the work [16] presents an effective analytical method for predicting the CTE of woven composites, and the results of comparison of prediction and experiment are given.

The works [37,38,39] are devoted to prediction of the CTE of composites using numerical methods and multiscale modeling. In these works, a satisfactory coincidence of the prediction results and the experimental data available in the literature is noted. The influence of the geometrical parameters of the composite structure determined by the reinforcing filler on the CTE was mainly investigated in these works.

Experimental studies of the temperature dependence of the CTE of polymers (including epoxy polymers) are presented in [11], where it is shown that the temperature expansion depends significantly not only on the temperature range, but also on the rate of temperature change; the presence of static stresses during cyclic tests (heating cooling) is also noted, which leads to a hysteresis effect on the curve of the relationship between the deformations and temperature.

From the practical and scientific points of view, the actual problem is the prediction of temperature dependence of GRP’s CTE based on experimentally obtained nonlinear temperature dependences of elastic characteristics and CTE of polymer matrices, as well as data on the geometry of composite structure. In this case, the results of prediction both before and after reaching the glass transition temperature of the binders are important.

Thus, the aim of this research is an experimental thermal expansion study of polymers and GRPs to realize the non-linear CTE model depending on heating temperature. It includes:-Thermosetting polymers’ thermal expansion experimental research before and after prolonged thermal aging.-Design and testing of non-linear analytic model of thermosetting polymers’ thermal expansion.-Experimental study of thermal expansion of GRPs with a matrix based on the previously studied thermosetting binders, before and after thermal aging.-Prediction of temperature dependences of the CTE of glass-reinforced plastics by means of finite-element homogenization and comparison with experimental results.

## 2. Materials and Methods

### 2.1. Materials

In this work, we consider glassed polymers made of epoxy, phenolic, epoxy-phenolic resins (presented in Table 1) and fiberglass plastics (presented in Table 2) made of epoxy and epoxy-phenolic resins and structural glass fabrics EZ-200 and T-23.

Epoxy binder (EP) for fiberglass plastic was made based on epoxy resin KER 828 (South Korea), which is an analog of the Russian resin ED-20, isomethyltetrahydrophthalic anhydride (ISOMTHFA) was used as a hardener, 2,4,6-tris-(dimethylaminomethyl)-phenol, produced under the brand name Alkophen was used as a curing gas pedal. The weight ratio of the ES components is as follows: KER 828-54.5%; IZOMTGFA—42.5%; Alcophene—3%. The components described below were used to make the binders:Epoxy resin KER 828, with the following main characteristics: Epoxy Group Content (EGC) 5308 mmol/kg, Epoxide Equivalent Weight (EEW) 188.5 g/eq, viscosity at 25 °C 12.7 Pa.s, HCl 116 mg/kg, and total chlorine 1011 mg/kg. Manufacturer: KUMHO P&B Chemicals, Gwangju, South Korea.Hardener for epoxy resin methyl tetrahydrophthalic anhydride with the following main characteristics: viscosity at 25 °C 63 Pa.s, anhydride content 42.4%, volatile fraction content 0.55%, and free acid 0.1%. Manufacturer: ASAMBLY Chemicals company Ltd., Nanjing, China.Alkofen (epoxy resin curing accelerator) with the following main characteristics: viscosity at 25 °C 150 Pa.s, molecular formula C15H27N3O, molecular weight 265, and amine value 600 mg KOH/g. Manufacturer: Epital JSC, Moscow, Russian Federation.

Resol phenolic resin SFRZ-309 with the following main characteristics: viscosity at 25 °C 700 MPa.s, not more than 20% (m/m) water, not more than 20% (m/m) free phenol.

The components were mixed in the above proportions at room temperature of about 25 °C. Mixing to a homogeneous consistency was carried out mechanically with an electric drill with a mixing attachment.

Glass cloth EZ-200 is produced according to the Russian standard GOST 19907-83 and has the following characteristics:-Thickness 0.190 +0.01/−0.02 mm;-Surface density 200 +16/−10 g/m^2^;-Number of yarns per 1 cm of fabric on the basis 12 +/− 1;-Number of yarns per 1 cm of fabric on the weft 8 +/− 1;-Weave–plain;-Oiling agent–paraffin emulsion.

Glass fabric T-23 is produced in accordance with Russian standard GOST 19170-2001 and has the following characteristics:-Thickness, 0.27 +0.01/−0.02 mm;-Surface density, 260 +25/−25 g/m^2^;-Number of yarns per 1 cm of fabric on the basis 12 +/− 1;-Number of yarns per 1 cm of fabric on the weft 8 +/− 1;-Weave–plain;-Oiling agent–aminosilane.

Samples of fiberglass plastic were made in the form of plates of 15 × 15 cm. Cut sheets of glass fabric EZ-200 were calcined at 300 °C to remove the paraffin oiling agent immediately before impregnation with the binder. Glass fabric T-23 was not calcined. In total, the samples had 10 layers of glass fabric laid according to the scheme 0/90 (base/weft).

Glass-reinforced plastic specimens were cured, at 120 °C, for 20 min in silicone molds, while being loaded through Teflon-coated metal plates at a pressure of about 0.22 kPa. The cured specimens were then kept at 150 °C for 12 h. After that, beam samples were cut from the plates in the direction of the main axes of orthotropy, which were considered in this work.

### 2.2. Methods

#### 2.2.1. Long Heat Treatment (Thermal Aging)

After curing, some of the fiberglass samples were exposed to prolonged exposure at elevated temperatures, while the control series was stored under normal conditions. The long-term curing (hereinafter referred to simply as “curing”) of the samples at elevated temperatures was performed according to the following program: 168 h (one week) at 160 °C, 168 h at 190 °C, 168 h at 220 °C. After the heat treatment, the samples were cooled at a rate of about 1 °C per minute to 50 °C, removed from the laboratory oven, weighed, and then weighed and tested for three-point bending at temperatures from 25 to 180 °C.

#### 2.2.2. Dilatometric Investigation

Dilatometric investigation of polymer and GRP samples (Figure 1) were performed with dilatometer Netzsch DIL 402 C (Figure 2) for CTE of solid materials determination. Testing was performed in argon.

Netzsch DIL 402 C technical characteristics:-Temperature range: 20–1500 °C-Colding and heating intensity: 0.01 K/min–50 K/min (5 K/min in experiment)-Etalon: Al_2_O_3;_-Linear range: 500 mcr;-Sample length: max. 28 mm;-Sample diameter: max. 12 mm;-Expanding Δl accuracy: 0.125 nm;-Atmosphere: inertial dynamic argon with gas flowing controller.

#### 2.2.3. Prediction of the Thermal Expansion Coefficient (CTE)

Prediction of the CTE of GRP as a function of temperature was performed in the academic version of the ANSYS finite element package using the built-in module Material Designer. Calculation of the elastic characteristics and CTE of the woven composite was performed based on the finite-element (FE) homogenization method [35,36]. In this method, a small representative volume of the material (representative volume element (RVE)) is extracted, while having sufficient dimensions to possess macroscopic characteristics of mechanical and thermomechanical properties.

The Material Designer module of Ansys Workbench automatically builds the bulk FE model (Figure 3) of the unit cell and calculates the orthotropic or fully anisotropic elastic characteristics of the homogenized material based on the specified material characteristics of the matrix and filler as a fabric. This method of modeling considers the curvature of the threads in the fabric structure as well as the influence of the transverse threads on the elastic modulus in the longitudinal direction.

The geometrical parameters of the RVE set as input data, as well as the boundary conditions and the procedure for calculating the elastic characteristics, are described in [40,41,42].

To calculate the CTE when the temperature changes by $\Delta T_{i}$ (the initial reference temperature in our calculations was taken as 23 °C), together with the mechanical boundary conditions, a uniform final temperature $\Delta T_{i}$ is applied to the model, and then the effective macroscopic RVE deformations and stresses in the Cartesian axis directions caused by temperature changes are calculated. The effective coefficient of thermal expansion can be calculated from the equation $\varepsilon_{it}=\alpha_{i}\Delta T_{i}$, where $\varepsilon_{it}$ is the increment of relative strain at the *i-th* step from the temperature change at $\Delta T_{i}\;$ relative to the base temperature $\Delta T_{i-1}$ at the *i-th* part of the temperature curve.

The RVE geometrical parameters were determined based on the study of the mesostructure features of fiberglass plastics using Levenhuk DTX 90 and Levenhuk 320 BASE optical microscopes equipped with digital cameras. The methodology for determining the geometric parameters of the structure of fiberglass plastics based on glass fabrics set in the Material Designer package is described in detail in [40,42].

Digital photos (Figure 4, Figure 5) of the structure of fiberglass samples were imported into the NanoCAD SPDS package (version for educational institutions) and brought to the same scale. The thickness of the sample at the marked point was taken as a reference dimension, which was measured with a caliper with an accuracy of 0.05 mm. After reduction to the same scale, the thickness of layers, shape and size of yarns, fractions of space occupied by yarns inside samples (Yarn volume fraction) and fractions of space occupied by fibers inside yarns (Yarn fiber volume fraction) were estimated. The RVE geometric model image from the Material Designer module was also imported and scaled for comparison with the real structure of the samples.

## 3. Results

### 3.1. Analytic Prediction

On the base of dilatometric testing of the glassed thermosetting polymers samples, the thermal expanding curves depending on heating temperature (to 200 °C) were obtained and presented in Figure 6. The main characteristic results are presented in Table 3. The following analysis of points on the curves specifies the polymers’ thermal expansion:The thermal expanding (TE) has non-linearity from normal to extreme temperature that probably accords to the glassing temperature of polymer Tg, and TE can be characterized by the coefficient of integral non-linear thermal expanding (CNTE) α_nl_;After temperature breaking point, the TE achieves the extremum and linearity of expansion with further heating; and can be characterized by coefficient of linear TE (CLTE) α_tg_;According to DMA results, the Tg is equal to temperature T_nl_, finishing the period of polymers’ thermal expanding non-linearity [22];Prolonged thermal relaxation (at aging) of epoxy and phenolic glassed polymers (GP) results in some reduction in epoxy CNTE and an increase the phenolic and epoxy-phenolic CNTE, thus increasing all polymers’ CLTE;CNTE changing after thermal relaxation is conditional, because CNTE depends on extreme non-linear relative expansion ε_nl-max_ that accords to temperature of GP: thermal relaxation causes an increase the ε_nl-max_ containing epoxy resin, and a reduction in the ε_nl-max_ containing phenolic resin.

It was realized the model of non-linear GP thermal expanding on the base of molecules torsion and their straightening under heating. For this the basic element was presented as an expanding spiral. The branch of spiral is under linear TE. The analyze of the model points that a spiral under free thermal expansion will follow the linear TE, determined by the CLTE of the branch’s material and slope angle. The non-linear TE of spiral can appear only in constrained tightened conditions of expanding in a “tunnel”, determined by the placement of adjoining molecules. The model is presented in Figure 7.

The primary factors for modeling are basic length of spiral branch as l and its CLTE as α; dimension of molecule along the spiral ace before and after heating as x и $x_{t}$; dimension of the molecule perpendicular to the spiral ace before and after heating as y and $y_{t}$, according to expansions during heating Δl, Δx, Δy; angles of branches slopes before and after heating as β и $\beta_{t}$, with heating temperature change Δt. Further equations of model’s deformations are
(1)$$\frac{\Delta l}{l}=\alpha \Delta t$$
(2)x=l×cosβ
(3)$$x_{t}=\left(l+\Delta l\right)\times cos\beta_{t}$$
(4)y=l×sinβ
(5)$$y_{t}=\left(l+\Delta l\right)\times sin\beta_{t}$$
(6)$$\frac{\Delta x}{x}=\frac{x_{t}-x}{x}=\frac{\left(l+\Delta l\right)\times cos\beta_{t}-l\times cos\beta }{l\times cos\beta }=\frac{\left(1+\alpha \Delta t\right)cos\beta_{t}}{cos\beta }-1$$

Under absolute spiral/molecule expansion that accords to extreme extensibility under heating and achieves linear TE section, $\beta_{t}$ = 0, $cos\beta_{t}=1$. Then
(7)$$\varepsilon_{nl-max}=\frac{1+\alpha_{tg}\Delta t}{cos\beta }-1$$
(8)$$\beta =arccos\frac{1+\alpha_{tg}\Delta t}{1+\varepsilon_{nl-max}}$$

As can be seen, the model provides non-linearity, but proposes the reduction in CNTE under heating, and that is opposite to the experimental results. Spiral extensae has the specify that linear expanding of the branch goes to the more ace expanding of spiral, with angle slope reducing ace expanding intensity reducing too. In practice, polymers’ behavior under heating is inverse, CNTE increases under heating. Instead of extension, the compression of branched molecules is going on.

Thus, the model of GP thermal expansion based on the previously designed domain supra-molecular models is proposed. The domain model includes two basic supra-molecular elements: rigid polymerized domain and ITZ, consisting of flexible and volatile organic molecules. In that case, the GP thermal expanding process under heating is presented as a simultaneous expanding of domain and compression of flexible ITZ elements until Tg, when flexible ITZ chains lose their elasticity absolutely and achieve the most possible compression. After Tg, the expansion of not bonded but directly relied on each other domains occurs. As a result, flexible ITZ molecules compensate deformations until Tg resulting in a non-linear character of GP expanding. The compensative model of GP expanding is presented in Figure 8.

To discover the mechanism of non-linear thermal expansion, figure the micro-composite two phase domain model in spherulitic type consisted of rigid linear TE spherulitic domains connected by spiral damping covers: TE develops along normal to domain surface (Figure 9). The compensation mechanism concludes in the process of radial expanding of the domain, that forces the tangential expansion of connecting spirals and compression along normal direction, compensating the total linear TE.

Then, the total radial expansion is the subtraction between the radial expansion of the domain and the normal compression of the spiral amplitude. In the moment that the extreme spiral compression is achieved, the domains begin to expand without compensation and polymer transverse to linear TE.

The primary factors for modelling are: domain radius as r; radius of supra-molecular element (SE) consisting of the domain and flexible molecules as R; spiral wave amplitude as A; radial expansion of domain Δr, of SE as ΔR, of amplitude as ΔA; tangential expansion of domain as Δx; total radial relative expansion as $\varepsilon_{r}$; total tangential expansion as $\varepsilon_{x}$; angle of the spiral branch slope before and after heating as β и $\beta_{t}$ with heating temperature change Δt.
(9)$$\varepsilon_{r}=\frac{\Delta r-\Delta A}{R}$$

Amplitude compression is influenced by tangential expansion of domain, so
(10)$$\Delta A=\Delta x\times tg\beta_{t}=\frac{\pi }{n}\Delta r\times tg\beta_{t}$$
with *n* equal to the frequency of the spiral wave.

We change the $\frac{\pi }{n}\times tg\beta_{t}$ на $tg\beta_{kt}$, with conditional angle of spirals branch slope $\beta_{kt}$. We take the CLTE of the domain equal to CLTE under Tg $\alpha_{tg}$. Then, the total relative expansion is
(11)$$\varepsilon_{r}=\frac{\Delta r-\Delta r\times tg\beta_{kt}}{R}=\frac{\Delta r}{R}-\frac{\Delta r}{R}\times tg\beta_{kt}=\alpha_{tg}\Delta t-\alpha_{tg}\Delta t\times tg\beta_{kt}$$
(12)$$\varepsilon_{r}=\varepsilon_{nl-max}\left(1-tg\beta_{kt}\right)=\alpha_{tg}\Delta t\left(1-tg\beta_{kt}\right)$$

CNTE with determined temperature is
(13)$$\alpha_{t}=\frac{\varepsilon_{nl-max}\left(1-tg\beta_{kt}\right)}{\Delta t}$$

This model determines the non-linear increasing of CTE under heating. With the branch slope angle reducing the total linear TE increases according to real polymers’ behavior. So,
(14)$$\beta_{kt}=arctg\left(1-\frac{\varepsilon_{r}}{\alpha_{tg}\Delta t}\right)=arctg\left(1-\frac{\alpha_{t}\Delta t}{\alpha_{tg}\Delta t}\right)=arctg\left(1-\frac{\alpha_{t}}{\alpha_{tg}}\right)$$

The spiral branches’ slope angle calculation with experimental expansions results in a change of the angle from 20 to 45 under heating, depending on polymer type. The thermal relaxation causes the increase in the slope angle, i.e., the tangential compression of the spiral cover grows. 

Modelling the CNTE in relation with temperature, beginning from the normal Tg = 20 °C, the wave equation was determined as
(15)$$tg\beta_{kt}=cos\frac{t-t_{0}}{t_{g}-t_{0}}=cos\frac{\Delta t}{\Delta t_{g}}$$

So, the CNTE model is
(16)$$\alpha_{t}=\frac{\varepsilon_{nl-max}\left(1-cos\frac{\Delta t}{{\Delta t}_{g}}\right)}{\Delta t}$$

In Figure 10, the experimental and model curves of GP CNTE are presented. It is obvious that the epoxy resin has more, and the phenolic resin has less CNTE and CLTE. Thermal relaxation goes to CNTE and CLTE reduction. It is related with structural changes in the polymer; increasing of Tg influence the increasEPEPing of total non-linear TE. The models calculations and experimental results’ deviation not more than 10% in 90% of cases. 

The interest is CNTE/CLTE temperature relationships of composites, for example, of compounds and GRP.

We present the total TE of the composite $\Delta L_{0}$ as the sum of the TEs of non-homogenic two-phase structures SEs $\Delta L_{1}$ и $\Delta L_{2}$.
(17)$$\Delta L_{0}=\Delta L_{1}+\Delta L_{2}$$

We determine the deformations as total ($\propto_{0}$) and element ($\propto_{1},\;\propto_{2},$) CLTE
(18)$$\Delta L_{0}=\propto_{0}*L_{0}*\Delta T,$$
(19)$$\Delta L_{1}=\propto_{1}*L_{1}*\Delta T,$$
(20)$$\Delta L_{2}=\propto_{2}*L_{2}*\Delta T$$

$L_{0}$, $L_{1}$, $L_{2},$—element dimensions of SE, flexible element and domain element.

We incept in (17) Equations (18)–(20)
(21)$$\propto_{0}*L_{0}*\Delta T=\propto_{1}*L_{1}*\Delta T+\propto_{2}*L_{2}*\Delta T,$$
(22)$$\propto_{0}*(L_{1}+L_{2})=\propto_{1}*L_{1}+\propto_{2}*L_{2}$$

With $k_{1}=\frac{L_{1}}{L_{1}+L_{2}}$, the equation becomes
(23)$$\propto_{0}=k_{1}\propto_{1}+k_{2}\propto_{2}=k_{1}\propto_{1}+(1-k_{1})\propto_{2}$$

Figure 11 presents the dilatometric curves of GRP with EP, EP-PF polymers and with two types of glass fiber before and after prolonged thermal relaxation.

Obviously, the GRPs’ CTE temperature curves are non-linear until the GP Tg. After Tg, GRPs almost lose the ability to expand due to the loss of bond between glass fiber and polymer. Aged at high temperature plastics keep some TE ability after Tg too. Significantly, the CNTE of GRP is several times less than the CNTE of the base polymers (Table 3). This indicates that glass fiber (GF) holds the TE of the matrix polymer. The prolonged thermal relaxation influences the decrease of GRPs’ CNTE, probably related with an increase in glass fiber/polymer adhesion strength and, consequently, an increased influence of GF on TE holding.

In Figure 12, GRPs’ CNTE model curves made by Equation (23) are presented, and the polymers’ CNTE $\propto_{p}$ was taken in there by Equation (16). Obviously, the non-linearity of GRPs TE is totally determined by non-linearity of used GPs’ TE.

Primary testing showed significant exceeding of the TE results. It indicates the significant changing of polymer properties forced by GF adhesion limited by influencing distance, called the radius of correlation. Due to that, the correlation coefficient $k_{c}$ depended on the polymer and the GF type was introduced
(24)$$\propto_{0}=k_{c}k_{gf}\propto_{gf}+(1-k_{c}k_{gf})\propto_{p}$$

With CLTE of GF $\propto_{1}$ = 5 × 10^–6^ W/(m°C), for GRP EP + EZ $k_{c}=1.2$, EP + T23 $k_{c}=1.27$.

The model provides adequate accuracy in GRPs’ CNTE calculation. The deviation is 2–20%, and with an average deviation not more than 10%, it provides adequateness of the proposed non-linear models, and it provides the highest accuracy achieved with EP composites.

### 3.2. Prediction by Finite Element Homogenization

An important problem is to predict the physical and mechanical properties of the composite based on the known physical and mechanical characteristics of its components and the structure formed by them. In the present work the problem of prediction of CTE of GRP based on the following initial data was considered:-Experimental dependences of CTE of polymer matrices on temperature;-Experimental temperature dependences of elastic moduli of polymer matrices, obtained in [7];-Characteristics of fiberglass plastic structure (relative fiber and matrix content, average distance between strands, layer thickness, etc.);-Constant CTE (5 × 10^–6^ °C ^−1^) and modulus of elasticity (73 MPa) of glass fibers, taken by default for the E-Glass material from the material base of the ANSYS Workbench package.

At this stage of the work, the types of glass-reinforced plastics based on epoxy and epoxy-phenolic binders and glass fabrics EZ-200 and T-23, whose characteristics are described in Section 2.1, were considered in Table 4. Geometric structure parameters for EZ-200 and T-23 glass fabrics presented in Table 5. Each of the considered fiberglass plastics was considered before and after thermal aging to which the samples were subjected according to the program described in Section 2.2.1.

The dependences of the CTE for the phenolic binder SFRZH-309 were also obtained, but samples of fiberglass plastics on phenolic binder could not be made qualitatively, so there are no experimental results for such fiberglass plastics in this work.

The experimental temperature dependences of the CTE of polymer binders were determined in this work by dilatometric studies according to the method described in Section 2.2. The CTE values in different temperature ranges were determined from the graphs shown in Figure 6 and Figure 11. Figure 13 shows an example of a graphical determination of the relative strain values at different temperatures for a non-temperature-cured epoxy binder (a) and a fiberglass plastic based on this binder and glass fabric T-23 (b). The curves for the other types of binders and fiberglass plastics are determined in a similar way.

The dilatometric studies were carried out during heating and cooling of the samples, and a pronounced hysteresis effect is observed on the heating and cooling curves for GRPs, which is absent for unreinforced binders. This effect is probably due to the structural features of fiberglass reinforcement. The CTE for GRPs was determined from the heating experimental curve, since it is also determined in the ANSYS Workbench Material Designer module during conditional heating.

The results of determining the temperature dependences of the CTE of unreinforced binders obtained by dilatometric tests are shown in Figure 14.

According to the results after thermal aging, the CTE of epoxy (EP) binder within the temperature range of 50–100 °C remained practically unchanged (the curves almost coincide), but at temperatures above 100 °C, decreased by 7–33%.

A similar effect is observed for the epoxy-phenolic binder—in the temperature range above 100 °C, the CTE reduction after thermal aging was 12–45%.

The CTE of phenolic (PF) binder at temperature 50 °C is 2–2.7 times lower than epoxy-phenolic (EP-PF) and epoxy (EP), respectively; with increasing of temperature up to 190 °C, this difference even further increases (3–4 times).

The results of the experimental determination of the temperature dependences of the CTE of GRPs obtained by dilatometric tests, as well as the results of the calculated prediction of the CTE for the same glass-reinforced plastics, are shown in Figure 15, Figure 16 and Figure 17, as well as in Figure 18, Figure 19 and Figure 20. In the first case, the graphs are plotted against the CTE values, as they were obtained in the Material Designer program; in the second case, the dependences are plotted against the averaged values. The averaged CTE values for each *n-th* temperature value were determined as arithmetic averages for *n − 1*, *n*, and *n + 1* temperature values.

The solid lines on Figure 15, Figure 16, Figure 17, Figure 18, Figure 19 and Figure 20 represent the experimental curves of the CTE dependence on temperature; the dashed lines represent the curves based on the results of calculations. The curves for GRPs not subjected to thermal aging are shown in blue, and the curves for glass-reinforced plastics after thermal aging are shown in orange. The numbers in the graphs (Figure 18, Figure 19 and Figure 20) indicate the ratio of the calculated CTE to the experimental value at a given temperature.

At temperatures above the glass transition temperature, there is a sharp change in the CTE of both binders and fiberglass plastics based on them, which can be clearly seen in the graphs. The closest CTE values before and after the glass transition temperature (and, consequently, the sharp change in properties) were not averaged between each other.

Having analyzed and compared the results of the experiment and prediction, the following conclusions can be made:The outlines of curves of CTE dependence on temperature, built based on calculations, are in good agreement with the outlines of the experimental curves in all cases except for the case of fiberglass on the epoxy-phenolic binder after curing (break point on the graph of the experimental graph is around 90 °C, and the calculated one is around 130 °C). To establish the reason for this, it is necessary to conduct additional tests on more samples.A satisfactory concurrence of the numerical results of the prognosis and the experiment (their relations are shown in the graphs on Figure 18, Figure 19 and Figure 20) for all considered cases is observed in the temperature range from 50 to 100 °C, after the glass transition temperature best coincidence of numerical values of CTE is obtained for glass-reinforced plastics on epoxy resin, which were not subjected to thermal aging.For glass-reinforced plastics on epoxy-phenolic binder at temperatures exceeding 100 °C, the predicted CTE values were significantly higher than the experimental ones (2.33–10.18 times). The reason for such a significant mismatch will have to be clarified in the future.

In general, it can be concluded that the considered method of prediction can be satisfactory for practical use, especially in the range of temperatures from 50 to 100 °C. It should be considered that in this work, the thermomechanical characteristics of glass fibers were taken based on reference data rather than based on experiment, which could introduce an additional error in the results of computational prediction.

## 4. Conclusions

In this study, the dilatometric tests, thermal expanding non-linearity analysis and modeling of GP and GRPs were realized.

GPs and GRPs TE research was demonstrated as a method of supra-molecular research. The method is very useful for researching the behavior of composite and hybrid materials because dilatometric research makes possible to determine the degree of interaction between components under stress–strain condition.

Experimental research proves that GPs and GRPs thermal expanding has a non-linearity under heating until glassing temperature Tg, achieving and transversing to pseudo-plastic condition. Under heating until Tg, the increase in CNTE is 50% and more. Prolonged thermal relaxation (TR) at high temperatures influences the reduction in the CNTE, concluding in an increase in extreme GP expanding until Tg, which also increases under TR.

Based on these specifics and the polymer molecule spiral model, the domain TE model with compensative covers was designed and tested. As a result, the wave equation of thermal expanding compensation was discovered. The models were successfully tested on GPs’ and GRPs’ TE.

The analysis of composites’ behavior under heating showed that providing a high degree of adhesion, the component with less CNTE dominates and determines the CNTE of the composite. Additionally, vice versa, when adhesion degree is low, the properties determine the most flexible and expandable element. For example, epoxy GP have the CNTE much higher than GF, but GRPs’ CNTE just 1.5…2 times higher than GF and 5…6 times less than GPs. In EP-PF without TR, the epoxy CNTE is dominate, and then TR dominates the phenol CNTE due to an increase in their adhesion. So, the matrix is the epoxy in this system, and the phenol is the reinforcing aggregate.

On the base of supra molecular modelling, a path is opened to discover the relationship between thermal deformations and inner stresses and, consequently, between elasticity and CTE, discovering the physical base of Barker’s coefficient [34].

In the results of this study, the compensative math models are designed. The models provide an accurate prediction of non-linear TE and CTE under determined temperatures, making possible a more accurate calculation of the stress–strain condition of GRP structures exploited under heating.

Additionally, the method of prediction by the FE homogenization can be satisfactory for practical use, especially in the range of temperatures from 50 to 100 °C. It should be considered that in this work, the thermomechanical characteristics of glass fibers were taken based on reference data rather than based on experiment, which could introduce an additional error in the results of computational prediction.

## Figures and Tables

**Figure 1 polymers-14-04281-f001:**
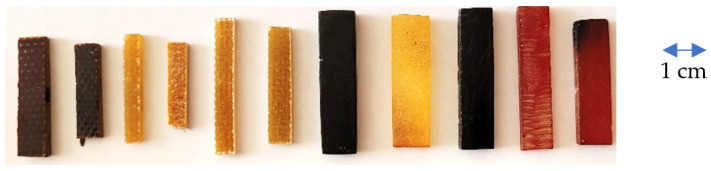
Examples of polymer and GRP samples.

**Figure 2 polymers-14-04281-f002:**
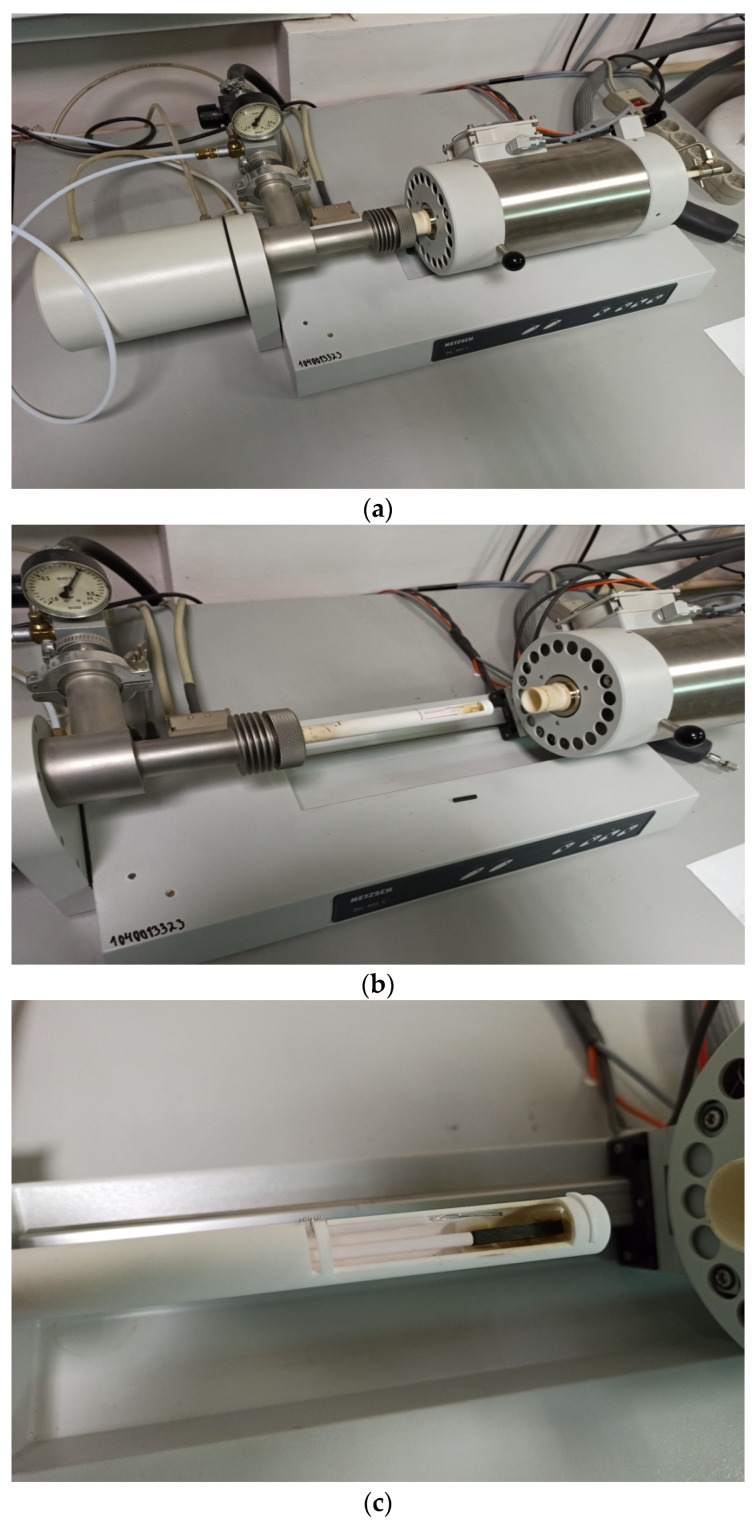
Dilatometer Netzsch DIL 402 C: (**а**) in process; (**b**) opened; (**c**) with sample (brown).

**Figure 3 polymers-14-04281-f003:**
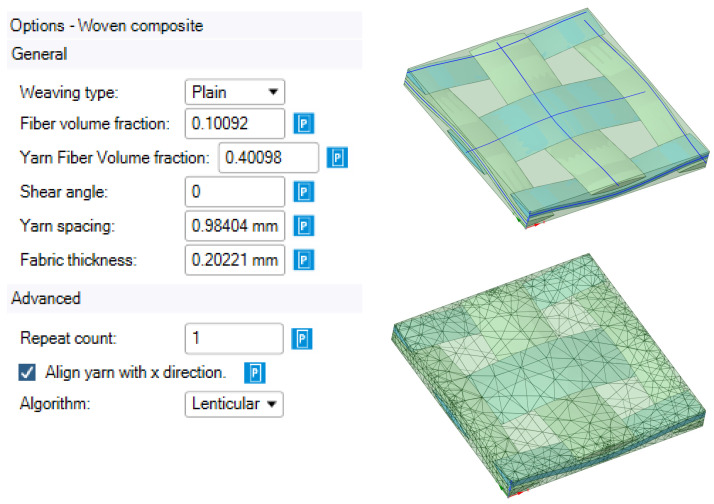
Initial data for constructing geometry, geometric and finite element RVE models.

**Figure 4 polymers-14-04281-f004:**
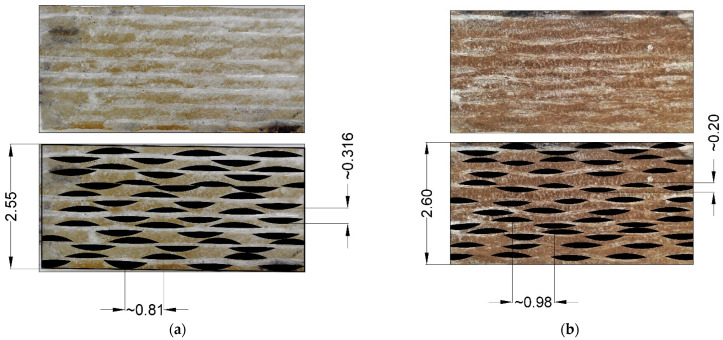
Microphotographs of the structure of fiberglass samples: (**a**) T-23; (**b**) EZ-200.

**Figure 5 polymers-14-04281-f005:**
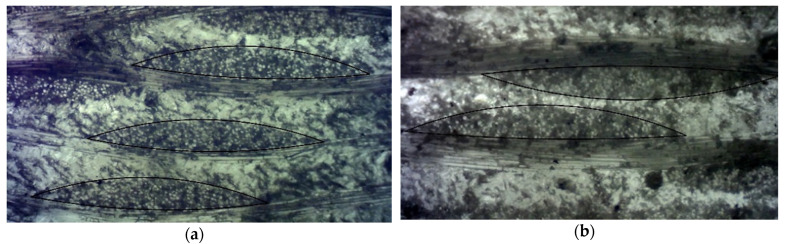
Microphotographs of filaments in the structure of fiberglass plastics with magnification 1:40: (**a**) T-23; (**b**) EZ-200.

**Figure 6 polymers-14-04281-f006:**
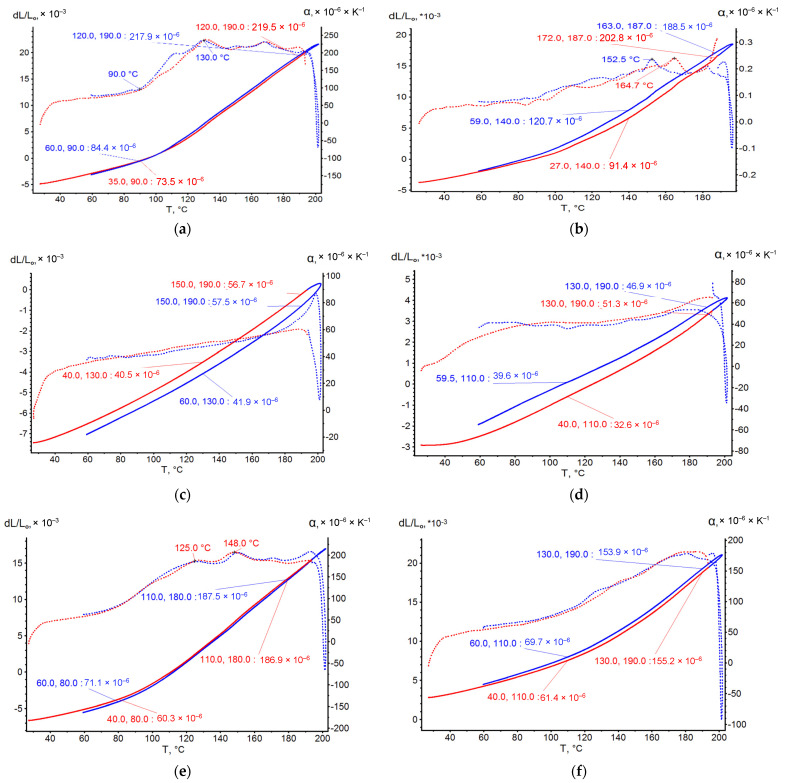
Thermal expanding curves: (**a**) EP, (**b**) EP after T–relax, (**c**) PF, (**d**) PF after T–relax, (**e**) EP–PF, (**f**) EP-PF after T-relax.

**Figure 7 polymers-14-04281-f007:**
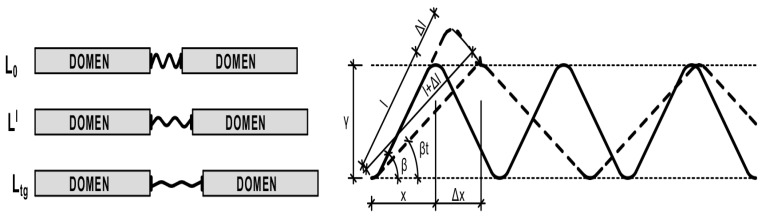
Model of polymer spiral expanding under heating.

**Figure 8 polymers-14-04281-f008:**
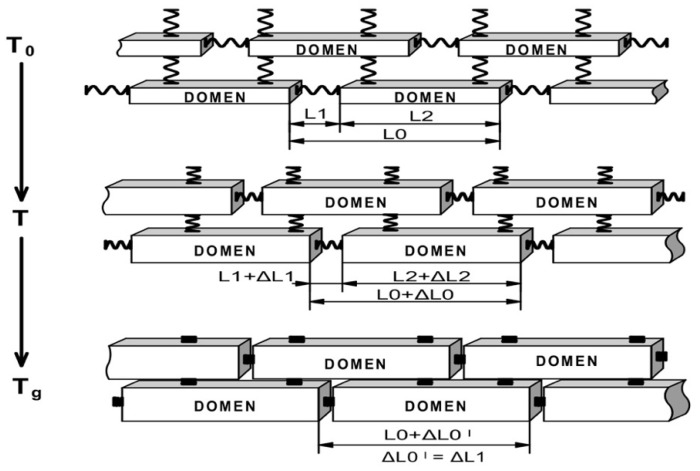
Compensative model of GP expanding under heating.

**Figure 9 polymers-14-04281-f009:**
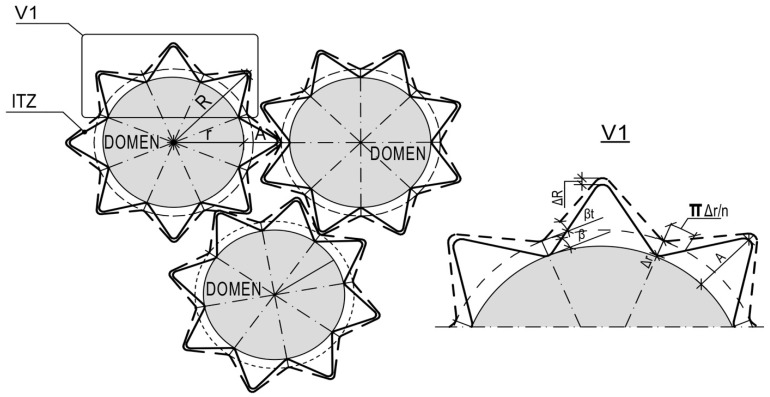
Domain spherulitic model of GP thermal expansion.

**Figure 10 polymers-14-04281-f010:**
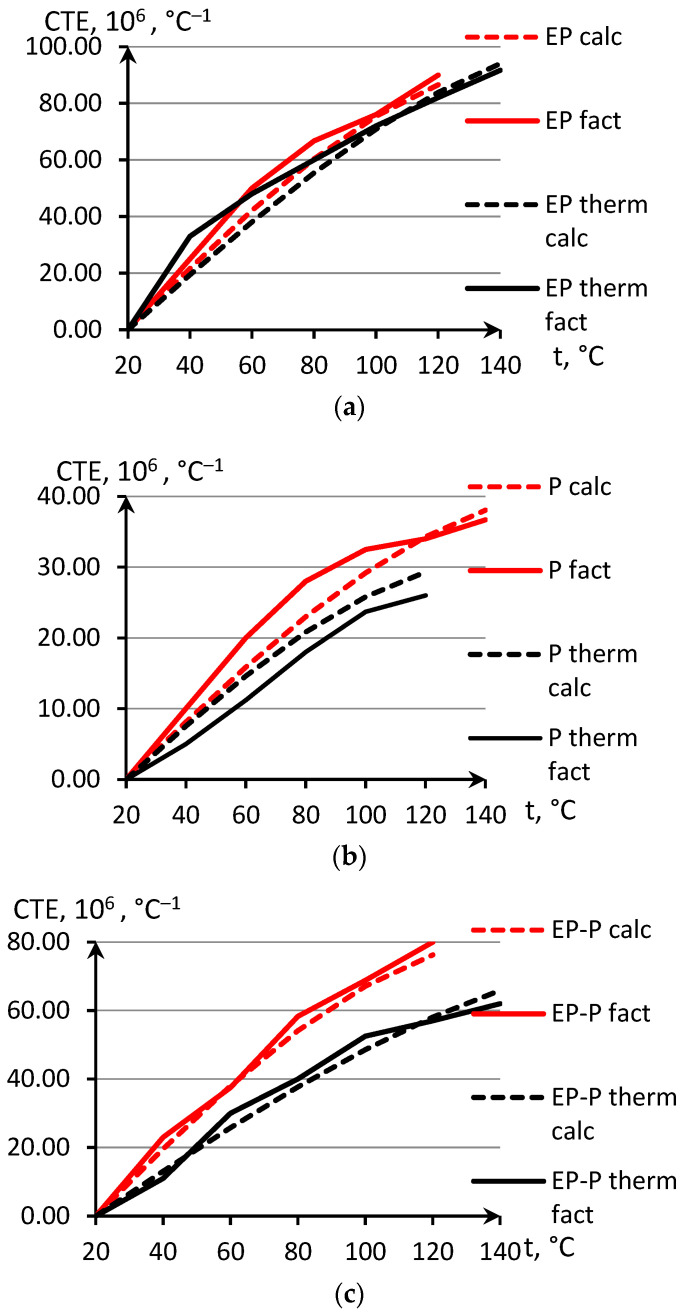
Experimental and model curves of GPs’ CNTE until T_g_.(**a**) EP, (**b**) PF, (**c**) EP-PF.

**Figure 11 polymers-14-04281-f011:**
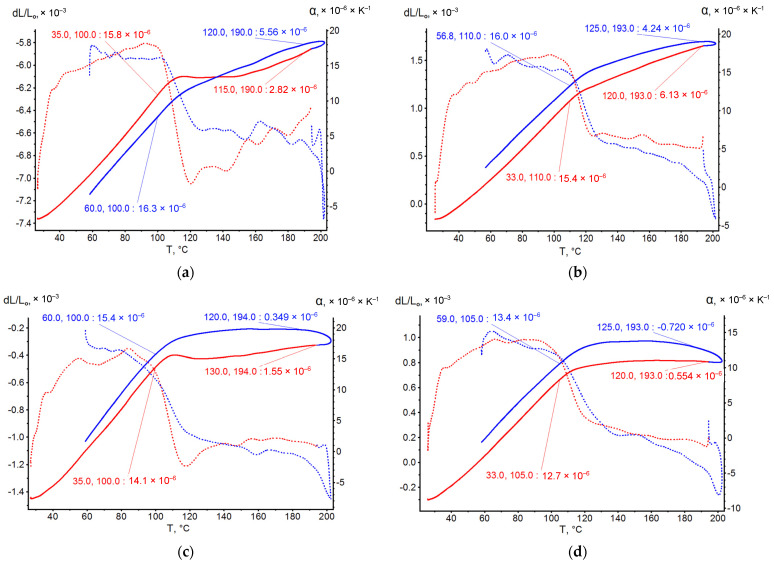
Dilatometric curves of GRPs under heating. (**a**) EP+EZ, (**b**) EP+EZ T-relax, (**c**) EP+T23, (**d**) EP+T23 T-relax, (**e**) EP-PF+EZ, (**f**) EP-PF+EZ T-relax.

**Figure 12 polymers-14-04281-f012:**
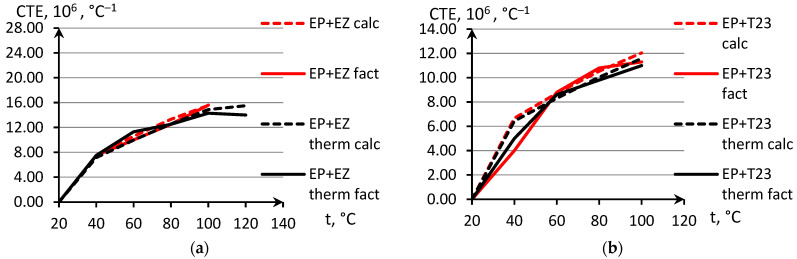
Experimental and model GRPs’ CNTE curves in temperature range of non-linear section. (**a**) EP+EZ, (**b**) EP+T23, (**c**) EP-PF+EZ.

**Figure 13 polymers-14-04281-f013:**
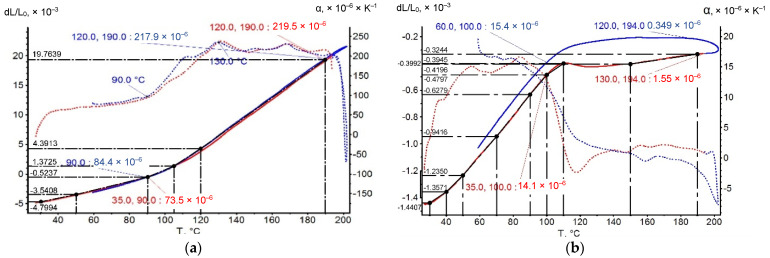
Examples of the graphical determination of the relative strain values for calculating the CTE: (**a**) for EP epoxy binder not subjected to thermal aging; (**b**) for fiberglass EP binder and T-23 fiberglass fabric.

**Figure 14 polymers-14-04281-f014:**
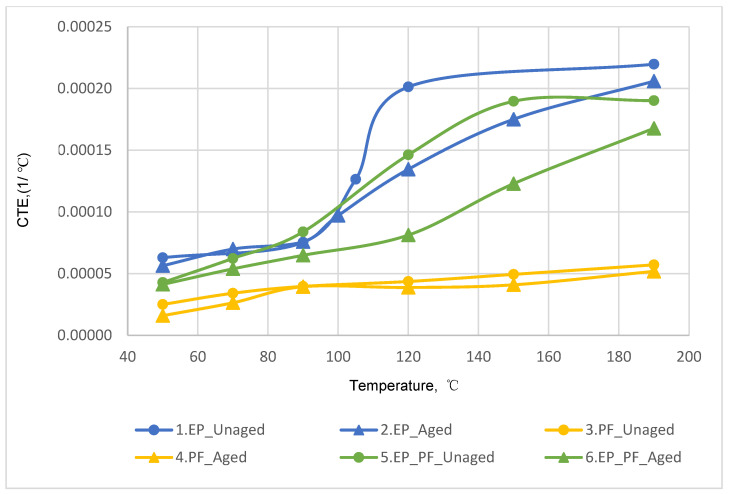
Temperature dependence curves for unreinforced binders based on dilatometry results: (1) epoxy binder without temperature aging; (2) epoxy binder after temperature aging; (3) phenolic binder without temperature aging; (4) phenolic binder after temperature aging; (5) epoxy-phenolic binder without temperature aging; (6) epoxy-phenolic binder after temperature aging.

**Figure 15 polymers-14-04281-f015:**
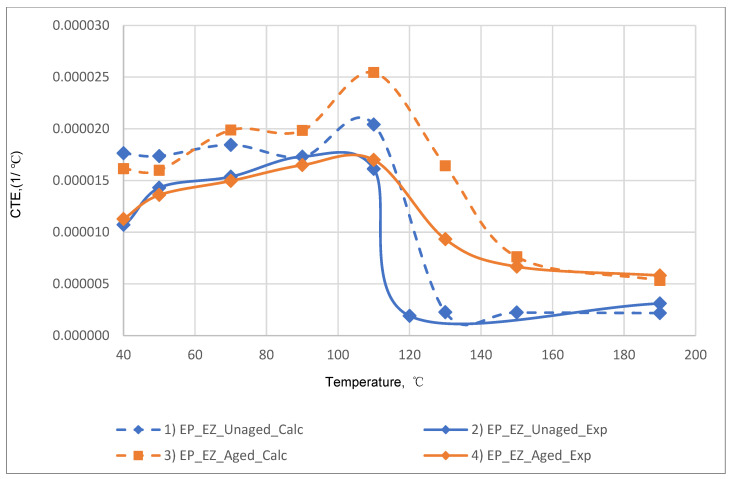
Dependence of CTE on temperature for epoxy fiberglass on glass fabric EZ-200: (1) calculated for EP-EZ200; (2) experimental for EP-EZ200; (3) calculated for EP-EZ200-TA; (4) experimental for EP-EZ200-TA.

**Figure 16 polymers-14-04281-f016:**
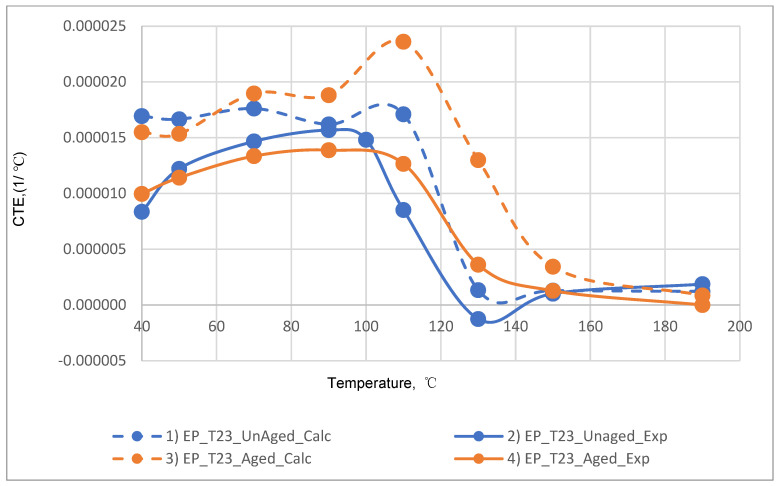
Dependence of CTE on temperature for epoxy fiberglass on fiberglass fabric T-23: (1) calculated for EP-T23; (2) experimental for EP-T23; (3) calculated for EP-T23-TA; (4) experimental for EP-T23-TA.

**Figure 17 polymers-14-04281-f017:**
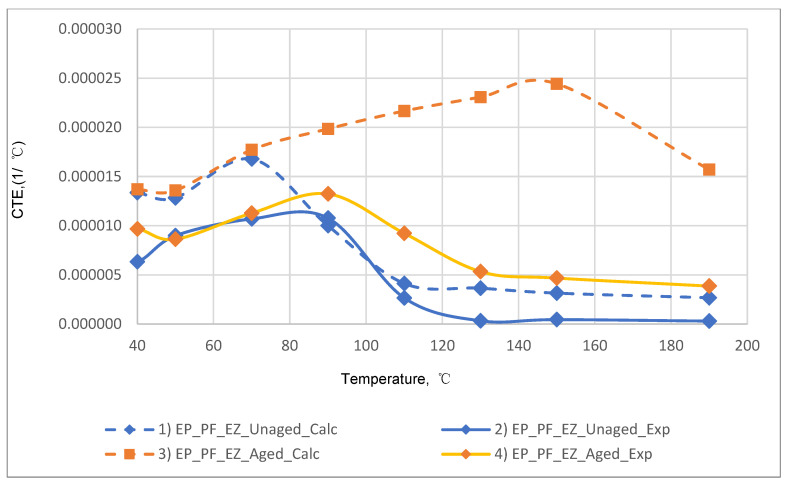
Temperature dependence of CTE for epoxy-phenolic fiberglass on EZ-200 fiberglass fabric: (1) calculated for EP-PF-EZ200; (2) experimental for EP-PF-EZ200; (3) calculated for EP-PF-EZ200-TA; (4) experimental for EP-PF-EZ200-TA.

**Figure 18 polymers-14-04281-f018:**
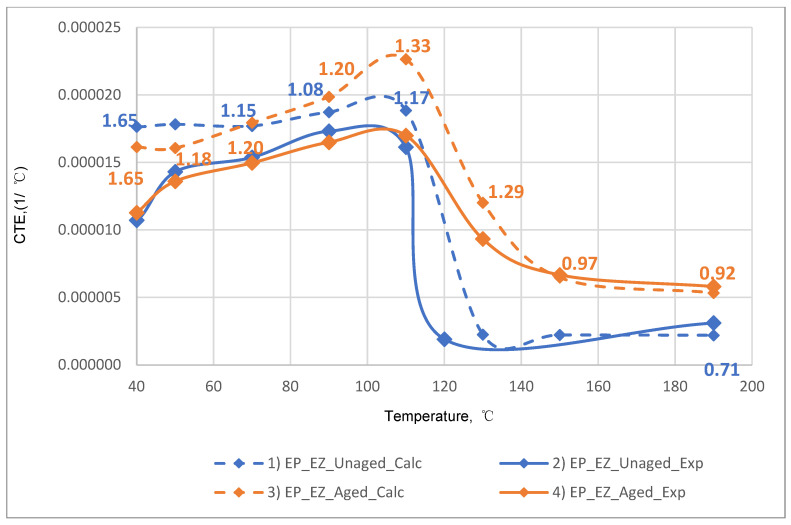
Dependence plotted against averaged CTE values on temperature for epoxy fiberglass on glass fabric EZ-200: (1) calculated for EP-EZ200; (2) experimental for EP-EZ200; (3) calculated for EP-EZ200-TA; (4) experimental for EP-EZ200-TA.

**Figure 19 polymers-14-04281-f019:**
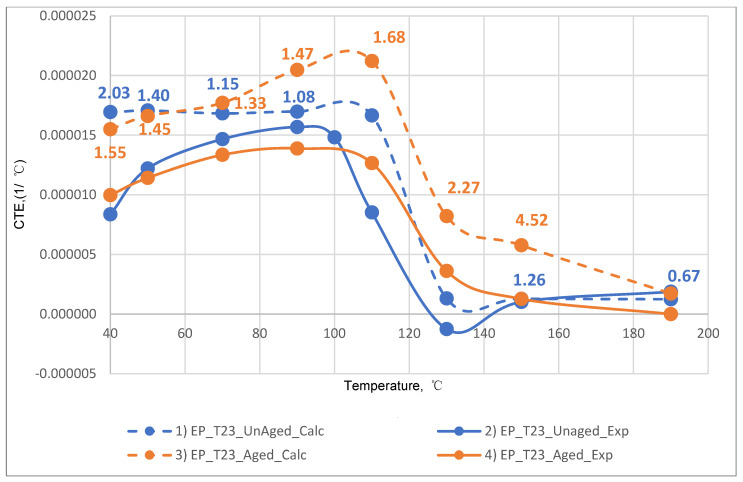
Dependence plotted from the averaged CTE values on temperature for epoxy fiberglass on glass fabric T-23: (1) calculated for EP-T23; (2) experimental for EP-T23; (3) calculated for EP-T23-TA; (4) experimental for EP-T23-TA.

**Figure 20 polymers-14-04281-f020:**
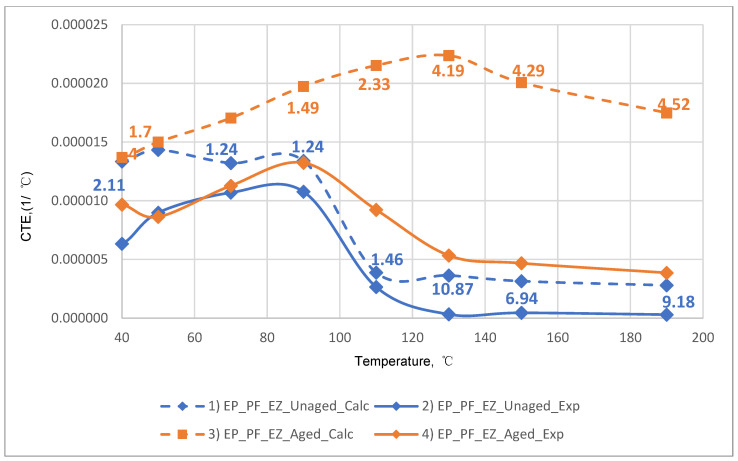
Dependence plotted on the averaged CCTE values on the temperature for epoxy-phenolic fiberglass plastic on glass fabric EZ-200: (1) calculated for EP-PF-EZ200; (2) experimental for EP-PF-EZ200; (3) calculated for EP-PF-EZ200-TA; (4) experimental for EP-PF-EZ200-TA.

**Table 1 polymers-14-04281-t001:** Types of binders investigated.

№	Composition	Name	Dilatometry
1	Epoxy (Ker 828 52.5% + MTHPA 44.5% + alkofen 3%)	EP	+
2	Phenolic (SFZ-309) 100%	PF	+
3	Epoxy-phenolic (ker 828 45% + SFZ-309 55%)	EP-PF	+

**Table 2 polymers-14-04281-t002:** Types of GRPs investigated.

№	Composite	Name	Dilatometry
1	EP 46% + EZ-200 54%	EP+EZ	+
2	EP 46% + T23 54%	EP+T23	+
3	EP-PF 46% + EZ-200 54%	EP-PF+EZ	+

**Table 3 polymers-14-04281-t003:** Dilatometry of GP results.

Composition	Tg, °C	T_nl_, °C	Mass Lost, %	Linear Shrinkage	Max α_nl_, 10^6^	α_tg_, 10^6^	ε_nl-max_
EP	115	115	-	-	88.7	219.5	0.0080
EP tr	136	138	0.058	0.018	91.4	202.8	0.0110
PF	243	130	-	-	40.5	56.7	0.0040
PF tr	-	110	0.072	0.022	32.6	51.3	0.0025
EP-PF	86	110	-	-	77.8	186.9	0.0065
EP-PF tr	136	150	0.06	0.019	76.6	155.2	0.0090

**Table 4 polymers-14-04281-t004:** Types of GRPs investigated.

№	Binder	Glass Fabric	Designation of Fiberglass	Thermal Aging
1	EP	EZ-200	EP-EZ200	–
2	EP	EZ-200	EP-EZ200-TA	+
3	EP	T-23	EP-T23	–
4	EP	T-23	EP-T23-TA	+
5	EP-PF	EZ-200	EP-PF-EZ	–
6	EP-PF	EZ-200	EP-PF-EZ200-TA	+

**Table 5 polymers-14-04281-t005:** Geometric structure parameters for EZ-200 and T-23 glass fabrics.

№	Type of Glass Fabric	Yarn Volume Fraction	Yarn Fiber Volume Fraction	Fiber Volume Fraction	Average Distance between the Yarns	Average Layer Thickness
1	T-23	0.3001	0.4088	0.122	0.81	0.316
2	EZ-200	0.2516	0.4009	0.101	0.98	0.202

## Data Availability

The data presented in this study are available on request from the corresponding author.

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
