# Peer review of "Non-Linearity of Thermosetting Polymers’ and GRPs’ Thermal Expanding: Experimental Study and Modeling"

_polymers, 2022, doi:10.3390/polym14204281_

Round 1
Reviewer 1 Report
In this paper, the nonlinear model of coefficient of thermal expansion (CTE) of polymer and FRP samples is discussed and verified. The analysis of the manuscript is somewhat superficial, and the expression needs to be further improved, so it is inevitable that there will be major changes. The following comments will help to improve the quality of this manuscript.
1. Abstract is the soul of a manuscript, and at least most of the content of the article should be condensed. The author does not provide the experimental methods and concluding information needed by interested readers in the abstract, which needs further revision.
2. The qualitative and quantitative analyses related to the test are also missing from the abstract. I hope the author can modify and supplement it.
3. There are a large number of text format errors in the paper, which makes it difficult to believe that the author has put his mind into the exposition of the thesis.
4. From Figure 1, it seems that the original dimensions of all the samples are inconsistent. The author should explain why there is such a consideration.
Author Response
Dear reviewer, thank you for your attention to our work and your objective comments, we have tried to make appropriate corrections and additions:
1, 2. The abstract has been revised.
3. we tried to correct all formatting errors in the text (local inconsistencies of fonts, captions under pictures, etc.)
4. With the dilatometric tests used, the dimensions of the specimens do not have to be the same, since before each test, the dimensions of each specimen are measured and entered as input data into the instrument test program. The difference in size was due to the fact that the specimens were machined for alignment.
Reviewer 2 Report
The manuscript is about experimental study and modeling of thermal expansion of thermoset polymers and FRPs. Minor revision is needed and the comments are listed below:
Introduction
Ø First paragraph, first sentence “The actual problem is ..…. formed by them”. It is quite strange to use this sentence as the opening of the introduction. It would be better to exchange the position of subject and object of this sentence.
Ø Second paragraph, “The most research …… hybrid materials”. Please give reference(s) to support this statement.
Author Response
Dear reviewer, thank you for your attention to our work and your objective comments, we have tried to make appropriate corrections and additions:
1. This paragraph has been reworded (lines 36...48).
2. References to works have been added (lines 49...50).
Round 2
Reviewer 1 Report
The latest manuscript indicates that the authors have revised the paper in detail and it now appears to be suitable for publication in this journal